# JAHS-Bench-201: A Foundation For Research On Joint Architecture And Hyperparameter Search

**Archit Bansal**[1]    **Danny Stoll**[1]    **Maciej Janowski**[1]    **Arber Zela**[1]    **Frank Hutter**[1,2]

[1]University of Freiburg    [2]Bosch Center for Artificial Intelligence

`{bansala,stolld,janowski,zelaa,fh}@cs.uni-freiburg.de`

## Abstract

The past few years have seen the development of many benchmarks for Neural Architecture Search (NAS), fueling rapid progress in NAS research. However, recent work, which shows that good hyperparameter settings can be more important than using the best architecture, calls for a shift in focus towards Joint Architecture and Hyperparameter Search (JAHS). Therefore, we present JAHS-Bench-201, the first collection of surrogate benchmarks for JAHS, built to also facilitate research on multi-objective, cost-aware and (multi) multi-fidelity optimization algorithms. To the best of our knowledge, JAHS-Bench-201 is based on the most extensive dataset of neural network performance data in the public domain. It is composed of approximately 161 million data points and 20 performance metrics for three deep learning tasks, while featuring a 14-dimensional search and fidelity space that extends the popular NAS-Bench-201 space. With JAHS-Bench-201, we hope to democratize research on JAHS and lower the barrier to entry of an extremely compute intensive field, e.g., by reducing the compute time to run a JAHS algorithm from 5 days to only a few seconds.

## 1   Introduction

In recent years, research on Automated Machine Learning (AutoML) [1] has made great strides in the data-driven design of neural network architectures [2, 3] and training hyperparameters [4]. Arguably, this fast pace of research on AutoML algorithms originates in the availability of several Neural Architecture Search (NAS) and Hyperparameter Optimization (HPO) benchmarks.

Tabular NAS benchmarks, such as NAS-Bench-101 [5] and NAS-Bench-201 [6], pre-evaluate all configurations in a given architecture search space and provide the results in a large table, which allows researchers to simulate runs of NAS algorithms in seconds on a single CPU-core that would have otherwise taken multiple days on a powerful GPU cluster. The availability of these benchmarks democratized NAS research, enabling small academic labs to work on NAS and allowing scientific comparisons between methods without confounding factors. The concept of tabular NAS benchmarks has also been extended to larger spaces by means of surrogate benchmarks, such as NAS-Bench-301 [7] and NAS-Bench-x11 [8].

While the availability of these cheap-to-evaluate NAS benchmarks has fueled the development of NAS algorithms (with more than 1000 papers on NAS in the last two years [9]), this push focused only on NAS in isolation, despite recent work showing that good hyperparameter settings can be more important than using the best architecture [10]. It would thus be much more powerful to treat these problems jointly, in what we thus call Joint Architecture and Hyperparameter Search (JAHS).

In order to facilitate work on JAHS, we build and release JAHS-Bench-201, an extended version of the popular NAS-Bench-201 architecture space that includes several hyperparameters and supports multi-fidelity, cost-aware and multi-objective optimization. Our individual contributions are:

36th Conference on Neural Information Processing Systems (NeurIPS 2022) Track on Datasets and Benchmarks.

Table 1: The parameters of our search (top and middle) and fidelity space (bottom) and how many unique values are available for a given discrete parameter.

| Space | Property | Description | # Values |
|---|---|---|---|
| Architecture | Cell Space | NAS-Bench-201 | 15,625 |
| Hyperparameter | Activation | ReLU/Hardswish/Mish | 3 |
| | Learning Rate | $[10^{-3}, 10^0]$ | Continuous |
| | Weight Decay | $[10^{-5}, 10^{-2}]$ | Continuous |
| | Trivial Augment | On/Off | 2 |
| Fidelity | N | Depth Multiplier | 3 |
| | W | Width Multiplier | 3 |
| | R | Resolution Multiplier | 3 |
| | Epoch | # Training Epochs | 200 |

- We created the most extensive dataset of neural network performance data available in the public domain, composed of approximately 161 million data points, each consisting of 20 performance metrics, across three image classification tasks, while featuring a 14-dimensional search and fidelity space (Section 2).

- Using surrogate models fit on our performance data, we derive benchmarking protocols covering different optimization tasks (three datasets, single/multiple objectives) and algorithmic settings (black-box, cost-aware, multi-fidelity, multi multi-fidelity). (Section 3)

- Our benchmark allows the study of several fundamental research questions in the JAHS domain, such as how important it is to optimize jointly in the space of architectures and hyperparameters, or how helpful different fidelity dimensions are (Section 4).

We provide access to our data and code to use our benchmarks alongside respective documentation at `https://github.com/automl/jahs_bench_201`. Finally, to further encourage research on JAHS, we maintain leaderboards for JAHS-Bench-201 at our aforementioned repository and provide first entries for them using simple baselines.

## 2 Design space, data collection, and surrogate modelling

We first describe the JAHS-Bench-201 design space of neural architectures, hyperparameters, and fidelity parameters. Then, we describe how we recorded 20 metrics for $270\,000$ configurations each on three image classification datasets and how we built surrogate models with this data.

### 2.1 Design space

The design space of JAHS-Bench-201 has two components: The search space and the fidelity space. The search space comprises the joint architecture and hyperparameter space of the JAHS problem, whereas the fidelity space comprises parameters that directly alter the training cost with some trade-off in performance. We include these fidelity parameters in our design space to support benchmarking of multi-fidelity algorithms [11–14] that use cheap, but low-fidelity, evaluations to gather knowledge quickly and thereby achieve speedups. Going beyond previous work in multi-fidelity optimization, we also introduce a *multi* multi-fidelity setting where the fidelity space has multiple (in our case four) dimensions that can be exploited.

**Search space**   We use the cell-based architecture search space from the popular and widely used tabular benchmark NAS-Bench-201 [6], which uses a directed acyclic graph with 4 vertices and 6 edges connecting them, for each of which there is a choice between an operation from the set of {skip-connect, zero, 1x1-conv, 3x3-conv, 3x3-avg-pool}. This accounts for a total of $5^6 = 15\,625$ possible architectures. See Appendix A for an illustration.

Our hyperparameter space consists of a mix of categorical and continuous hyperparameters. As categorical hyperparameters, we added a choice between 3 activation functions (Mish [15], Hardswish [16] or ReLU [17]) in the convolution layers, as well as the choice to enable or disable

Table 2: Overview of the image classification datasets we use in JAHS-Bench-201. "Base Image Size" refers to the image size corresponding to the highest fidelity we use.

|  | CIFAR-10 | Colorectal-Histology | Fashion-MNIST |
|---|---|---|---|
| # Images | 60,000 | 5,000 | 70,000 |
| # Classes | 10 | 8 | 10 |
| Original Image Size | 32x32x3 | 150x150x3 | 28x28x1 |
| Train-Valid-Test split | 40-40-20 | 63-27-10 | 63-27-10 |
| Base Image Size | 32x32x3 | 32x32x3 | 32x32x1 |

the use of Trivial Augment [18] for data augmentation in the training pipeline. For continuous hyperparameters, we added the choice of learning rate and weight decay. See Table 1 for a summary of the full search space. The original NAS-Bench-201 search space used ReLU activations only, did not use Trivial Augment, and used fixed values for learning rate and weight decay.

**Fidelity space** To support benchmarking of multi-fidelity algorithms [11–14], we consider four fidelity parameters that alter the training cost and performance: two architectural parameters controlling the network's modelling capacity, one to control the size of the input, and finally, the number of training epochs. We parameterised the cell architecture in order to represent two inherent properties of a neural network's modelling capacity - the network *depth* and the network *width*. Each cell may be repeated $N \in \{1, 3, 5\}$ times within a single block of the neural architecture. We call $N$ the *depth-multiplier*. The first convolution layer of any cell contains $W \in \{4, 8, 16\}$ filters, with the number of filters doubling in every subsequent convolution layer in the same cell. Thus, we call $W$ the *width-multiplier*. We note that, as in NAS-Bench-201, a single model contains only a single cell architecture. Additionally, to represent the impact of performing the same task on images of different sizes, we scaled the input images to different resolutions beforehand. This is denoted by the the *resolution-multiplier* $R \in \{0.25, 0.5, 1.0\}$. Here, $R$ is the fraction of the *base image resolution* that all inputs were scaled to before training, where base image resolution refers to the maximum square image size we use, which may or may not correspond to the native size of the images in that dataset (see Table 2). Since we recorded metrics for every epoch of training and evaluation, we can query the performance of every configuration at any epoch up to epoch 200. Therefore, we can use the number of epochs as a fidelity parameter, $epoch \in \{1, 2, \ldots, 200\}$. This number of epochs is indeed one of the most commonly used fidelity parameters in multi-fidelity optimization [14, 19].

## 2.2 Data collection and recorded metrics

We used three image classification tasks: CIFAR-10 [20], Colorectal-Histology [21] and Fashion-MNIST [22] (see Table 2), and for each task we randomly sampled 270 000 configurations from the design space and recorded 20 metrics at each epoch for every configuration, for both training and evaluation. As we ran 200 epochs of training, we, therefore, collected more than 161 million data points (see calculation below). In Section 2.3 we describe how we use the collected data to create surrogate models for JAHS-Bench-201. The performance metrics consist of values, such as the average runtime requirements for a single forward or backward pass per minibatch, the model loss and accuracy on each data split, *etc.* (We summarize them in Table 3.) See Appendix B for a full description of our training pipeline, including the fixed hyperparameters.

The benchmark dataset is thus queryable at any of the 200 epochs for any of these metrics, over the $3 \times 3 \times 3 = 27$ unique groups of values of the other three fidelity parameters (hereafter referred to as *fidelity groups*), and for up to 10 000 configurations over all the fidelity groups. Some configurations diverged before reaching 200 epochs of training; for these configurations, we only include the data until divergence. Combining three image classification tasks, 270 000 configurations per task and 200 data points per configuration amounts to a theoretical maximum of $162 \times 10^6$ possible data points (ignoring divergent configurations) and our dataset currently contains approximately $161 \times 10^6$ (161 million). We also discovered significant noise in the performance dataset for the Colorectal-Histology task and discuss this further in Appendix F.

Table 3: Summary of all metrics recorded in the benchmark dataset. The "Duration" metrics correspond to the average per-minibatch runtime cost of each epoch, "Loss" corresponds to the average Cross-Entropy loss and "Accuracy" corresponds to top-1 classification accuracy of each epoch. "FLOPS" and "Size" are calculated for the model as a whole. "Latency" is calculated as a per-datapoint running average every epoch. "Runtime" is the total cumulative time at each epoch that training and evaluation took, including any overheads not accounted for in "Duration".

| Metric | Splits | Description |
|---|---|---|
| Duration | Train/Valid/Test | Sum of the components (seconds) |
| Load data | Train/Valid/Test | Mini-batch transfer overheads within system memory |
| Forward | Train/Valid/Test | Forward pass duration |
| Backprop | Train | Backward pass duration |
| Loss | Train/Valid/Test | Loss metric value |
| Accuracy | Train/Valid/Test | Top-1 accuracy (percentages) |
| FLOPS | - | Floating Point Operations per Second |
| Latency | - | Average model latency |
| Runtime | - | Total runtime, including all overheads |
| Size | - | Estimated model size (MB) |

## 2.3 Surrogate models

As is common in benchmarking for Neural Architecture Search and Hyperparameter Optimization, to expand the scope of queries that our benchmark supports beyond just the samples present in the performance dataset, we utilize surrogate models fit on our collected data [7, 8, 23]. Based on the excellent results of surrogate models based on gradient boosting for both NAS benchmarks [24] and HPO benchmarks [23], we chose to use XGBoost [25] with optimized hyperparameters. We optimize over the same hyperparameter space as Mehta et al. [24] and use Bayesian optimization [26] with expert priors [27] as implemented in the NePS python package [28]. We trained a separate model for each metric. To validate our surrogate models, we again follow Mehta et al. [24] and list Kendall Rank Correlation Coefficients (Kendall's $\tau$) and Coefficient of Determination ($R^2$) scores in Table 4. We generally observe high scores; for CIFAR-10, for instance, Kendall's $\tau$ values up to 0.994 and $R^2$ scores as high as 0.998. Since a subset of the metrics present in the performance dataset can be subsumed by others, we chose to predict only 7 of the 20 available metrics. We list further details, such as how the training, validation and testing splits were generated, in Appendix C.

Table 4: We report the Kendall's $\tau$ correlation values and $R^2$ regression scores for the final surrogate models that we trained, for all 3 tasks and all 7 metrics for which surrogates were trained.

| | CIFAR-10 | | Colorectal-Histology | | Fashion-MNIST | |
|---|---|---|---|---|---|---|
| Metric | Kendall's $\tau$ | $R^2$ | Kendall's $\tau$ | $R^2$ | Kendall's $\tau$ | $R^2$ |
| FLOPS | 0.994 | 0.997 | 0.836 | 0.609 | 0.997 | 0.999 |
| Latency | 0.944 | 0.690 | 0.785 | 0.567 | 0.977 | 0.977 |
| Runtime | 0.954 | 0.907 | 0.725 | 0.556 | 0.983 | 0.992 |
| Size (MB) | 0.970 | 0.998 | 0.992 | 1.000 | 0.994 | 0.999 |
| Test-Accuracy | 0.892 | 0.953 | 0.691 | 0.772 | 0.907 | 0.953 |
| Train-Accuracy | 0.913 | 0.967 | 0.761 | 0.799 | 0.977 | 0.986 |
| Valid-Accuracy | 0.890 | 0.953 | 0.682 | 0.766 | 0.922 | 0.957 |

## 3 JAHS-Bench-201

Here, we use the design space and surrogate models introduced in Section 2 to define the benchmarks and corresponding evaluation settings that make up JAHS-Bench-201. We provide code to use JAHS-Bench-201 and follow our evaluation protocols alongside documentation at `https://github.com/automl/jahs_bench_201`. For all benchmarks and evaluation settings,

Table 5: A comparison of the compute requirements of surrogate-based experiments (values reported in CPU-seconds) vs training-based experiments (values reported in GPU-days).

| Task | Surrogate-based [**CPU-s**] | Training-based [**GPU-d**] |
|------|------|------|
| CIFAR-10 | 2.8 | 5.0 |
| Colorectal-Histology | 2.9 | 0.4 |
| Fashion-MNIST | 2.7 | 2.5 |

we maintain leaderboards as described in our code repository and provide first entries with random search and Hyperband [29].

**Optimization tasks**   We consider six optimization tasks in JAHS-Bench-201. For each of the three datasets (Table 2) we consider single-objective optimization (maximizing validation accuracy) and multi-objective optimization (finding the best trade-offs between high validation accuracy and low latency). While our surrogate models and our code allow for even more tasks, here, we focus on the most important ones that we also include in the JAHS-Bench-201 evaluation protocol below.

**Algorithmic settings**   JAHS-Bench-201 supports benchmarking in most algorithmic settings, however, we explicitly take into account the following four classes of algorithms that we maintain leaderboards for: black-box algorithms [30–32], cost-aware algorithms [33], multi-fidelity algorithms [14, 34, 35], and multi multi-fidelity algorithms [12]. Therefore, with 4 algorithmic settings and 6 optimization tasks, we maintain a total of 24 leaderboards.

**Evaluation protocol**   We accompany JAHS-Bench-201 with evaluation protocols and code to aid following these protocols, to facilitate fair, reproducible, and methodologically sound comparisons. Studies with JAHS-Bench-201 should report on all three datasets and present results as trajectories of best validation error (single-objective) or hypervolume (multi-objective). To allow using the same evaluation protocol and comparisons across the various algorithmic settings above, these trajectories should be reported across total runtime taking into account the training and evaluation costs predicted by the surrogate benchmark. We suggest to report until a runtime corresponding to approximately 100 evaluations (see Appendix D) and, for interpretability, show the total runtime divided by the mean runtime of one evaluation. Further, in (multi) multi-fidelity runs utilizing epochs, we support both using continuations from few to many epochs (to simulate the checkpointing and continuation of the model) as well as retraining from scratch for optimizers that do not support continuations. Results should report the mean and standard error around the mean and feature a minimum of 10 seeds.

**Runtime comparison**   In Table 5, we compare the CPU-runtime of a surrogate-based versus training-based run of random search, averaged across 100 different seeds, reiterating the speed-ups achievable by utilizing a surrogate model.

## 4   Experiments

In this section, we use JAHS-Bench-201 and the evaluation protocol in Section 3 to tackle three research questions about the Joint Architecture and Hyperparameter Search (JAHS) problem. We provide experimental details in Appendix D.

**Is it helpful to jointly optimize architecture and hyperparameters?**   In our first experiment, we explore the motivation behind JAHS and compare two algorithms, random search and Bayesian Optimization (BO) [31], on the joint space to an HPO space and a NAS space. We construct the latter two by constraining our JAHS space, hereby calling the derived spaces "HPO-only" and "NAS-only", respectively. Specifically, for NAS-only, we need to decide on a fixed hyperparameter setting (as any NAS would have to); to avoid introducing any bias by this choice, we sample a new random hyperparameter setting for each seed of NAS-only. Analogously, in the case of HPO-only, we sample a random fixed architecture for each seed. We find that JAHS outperforms HPO-only and NAS-only in terms of both anytime and final performance across all datasets (Figure 1 and Figure 2), sometimes with very substantial differences. We observe that, when using random search, for instance, in order to produce configurations that achieve a similar level of accuracy on a given task, JAHS can be expected to be 33 times faster than HPO-only and 29 times faster than NAS-only; see Table 9

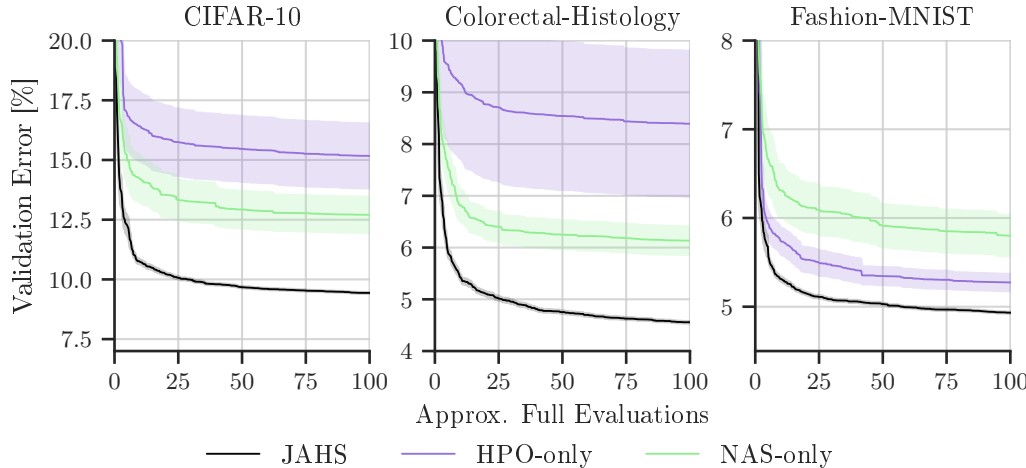

Figure 1: Mean validation error of random search applied to a NAS space, an HPO space, and the joint space (JAHS). Error bounds correspond to the standard error across 100 seeds. For each seed of NAS-only we sample fixed hyperparameter settings randomly, and, similarly, for HPO-only a fixed architecture.

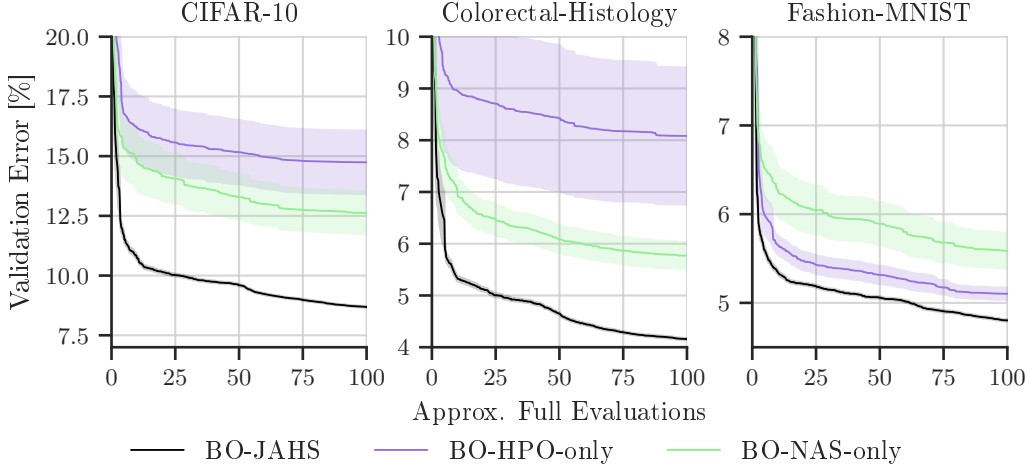

Figure 2: Mean validation error of Bayesian Optimization (BO) applied to a NAS space, an HPO space, and the joint space (JAHS). Error bounds correspond to the standard error across 100 seeds. For each seed of NAS-only we sample fixed hyperparameter settings randomly, and, similarly, for HPO-only a fixed architecture.

and Table 10 for exact performance speedups. JAHS also exhibits lower noise, indicating more robustness over HPO-only and NAS-only. Therefore, the answer to our research question is: *Yes, in the JAHS-Bench-201 setting it is helpful to optimize the architecture and its hyperparameters jointly.*

**Are some fidelities more useful than others?** While multi-fidelity optimization promises speedups, it is often unclear which fidelity parameter to choose for deep learning to achieve the largest speedups, with the number of epochs as a natural default choice. To compare the impact of different fidelity parameters, we run a common multi-fidelity algorithm, Hyperband [29] and its model-based implementation in SMAC [31], once for each of the four fidelity parameters in our design space. To set a baseline for *multi* multi-fidelity optimization, we run Hyperband traversing on the fidelity hypercube diagonal (Figure 3 and Figure 4). We find that the different individual fidelity parameters lead to indistinguishable results on CIFAR-10 and Fashion-MNIST. On Colorectal-Histology, the

width-multiplier performs worst in the model-free search, whereas the depth-multiplier performs best in model-based search. We find that diagonal traversal in the fidelity space does not bring speedups over single-fidelity optimization. Moreover, we find that in model-based search, diagonal traversal may even hinder the optimization (Figure 4). The answer to our research question is thus: *All four choices of fidelity perform similarly well, with the width multiplier performing worse in one case.* Since the default choice of number of epochs is not the only choice of fidelity that performs well, we expect that future work exploiting several of the fidelity choices at once in a more sophisticated *multi* multi-fidelity approach may lead to substantial further speedups.

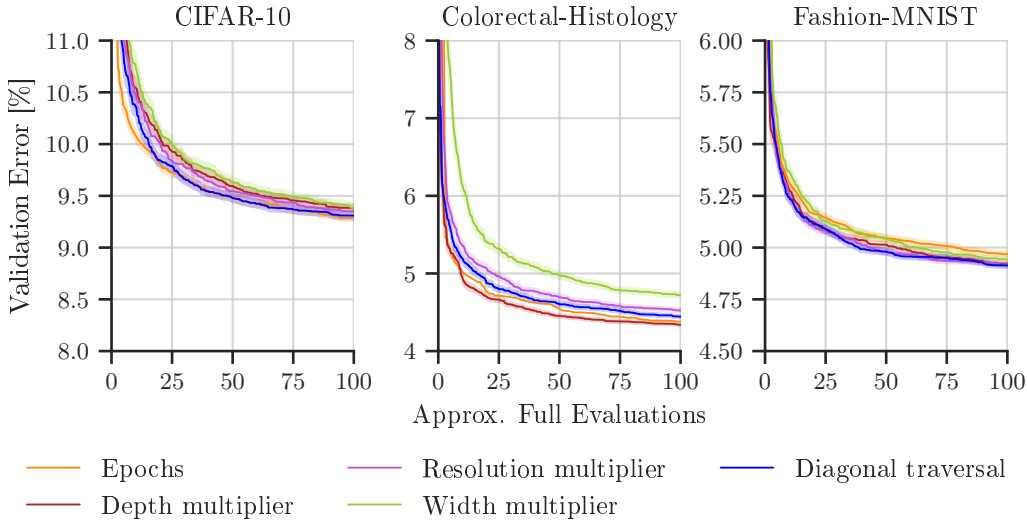

Figure 3: Mean validation error of Hyperband for the different fidelity parameters in JAHS-Bench-201. Error bounds correspond to the standard error across 100 seeds
.

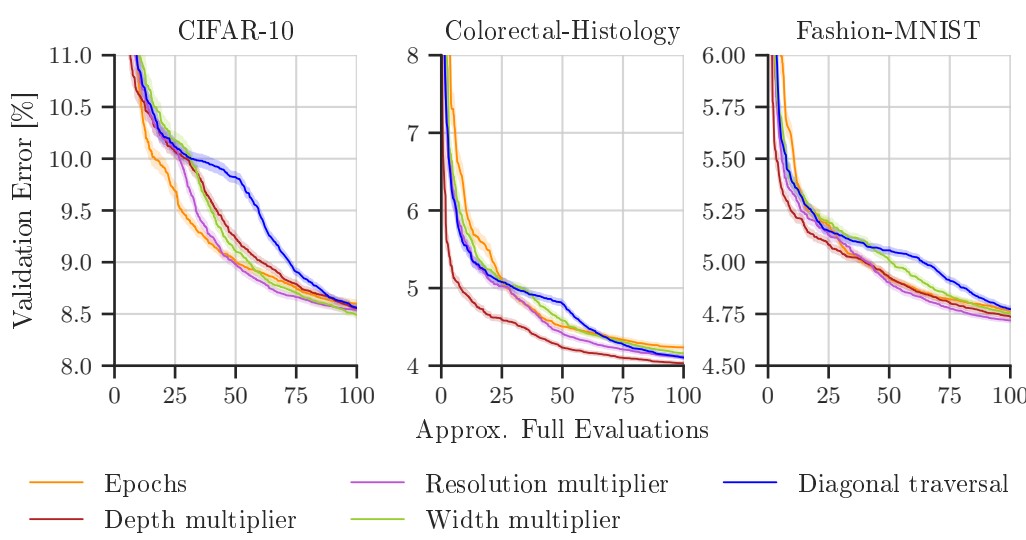

Figure 4: Mean validation error of model-based Hyperband for the different fidelity parameters in JAHS-Bench-201. Error bounds correspond to the standard error across 100 seeds.

**How correlated are the various metrics we collected to each other?** Even though we collect a large number of metrics, a number of them are intuitively closely related to each other. In order to

study this empirically, we examine the Kendall's tau rank correlation of our performance metrics and clearly observe that our metrics can be broadly clustered into two groups of highly correlated metrics (Figure 5).

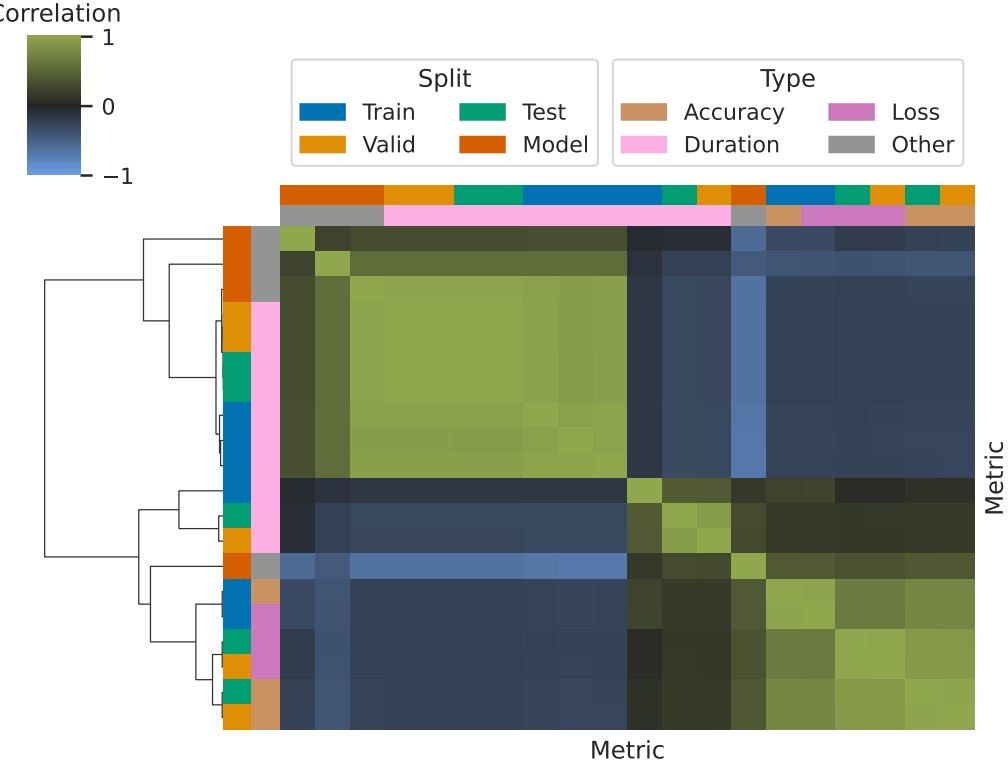

Figure 5: Kendall's tau rank correlations between different metrics for the task CIFAR-10. Each row and column corresponds to one of the metrics we collected, represented using two sets of color codes for ease of visualization. The codes for "split" represent if a metric is associated with one of the sub-splits of the task or the model as a whole, whereas the "type" codes represent the type of metric from one of 4 broad categories that our metrics were divided into. The dendrogram on the left indicates a grouping of the rows based on similarity of the entire row.

## 5 Related benchmarks

While there already exists a long line of research that tackles easy and accessible benchmarking in NAS [5, 6, 36, 37, 8, 38–40] and HPO [41, 23, 42, 43] [44] separately, benchmarking in the joint NAS and HPO space has not been extensively adopted by the community yet, mainly due to the scarcity of algorithms that operate in such a joint space. In contrast, various pure special-case NAS benchmarks have been constructed by the NAS community due to the increasing popularity of NAS [45]; these tackle different tasks and spaces, such as NAS-Bench-ASR [39] for speech recognition, NAS-Bench-NLP [38] for natural language processing, HW-NAS-Bench [46] for hardware-aware NAS, TransNAS-Bench-101 [47] and NAS-Bench-360 [48] for a multitude of diverse tasks and spaces, and NAS-Bench-x11 [8] for learning curve predictions.

Despite the scarcity of JAHS algorithms, there already exists first work on benchmarks in this domain. NAS-HPO-Bench [49], LCBench [50] and NAS-HPO-Bench-II [51] are notable forerunners of JAHS-Bench-201. However, our work offers a broader support compared to these benchmarks in many dimensions, such as multi-fidelity and multi-objective support, the broader nature of the hyperparameter space, and the number of stored evaluations in the benchmark (with ours being roughly 700 times larger than the previous largest one, NAS-HPO-Bench-II). In Table 6, we provide a comparison to other benchmarks (including some prominent pure NAS benchmarks) to demonstrate their similarities and differences with our work. We note that, compared to NAS-Bench-HPO

Table 6: A broad comparison of some of the features of benchmarks that are very closely related to ours. For brevity, we used these abbreviations: JAHS-Bench-201 → Ours, NAS-HPO-Bench → NHB, NAS-HPO-Bench-II → NHBII, NAS-Bench-101 → NB101, NAS-Bench-201 → NB201, NAS-Bench-301 → NB301, NATS-Bench → NATSB. "Size Arch. Space" and "Size HP Space" refer to how many unique values are possible within the architecture and hyperparameter search spaces, respectively. We use these symbols: (†) only the topology space is considered here, the size space [40] is considered the fidelity space; (‡) details on the number of seeds were not given, thus, this is only the number of available architectures; (∗) epochs is supported, but not full learning curves; (§) only the size search space provides multi-fidelity support.

| | Ours | NHB | NHBII | NB101 | NB201 | NB301 | NATSB |
|---|---|---|---|---|---|---|---|
| Size Arch. Space | 15.6K | 144 | 4K | 423K | 15.6K | $10^{18}$ | 15.6K$^\dagger$ |
| Size HP Space | Continuous | 432 | 48 | - | - | - | - |
| # Evaluations | 161M | 62K | 192K | 5M | 33K$^\ddagger$ | 60K | 145K$^\ddagger$ |
| # Metrics | 20 | 7 | 3 | 5 | 4 | 9 | 11 |
| Multi-Fidelity? | Yes | Yes | Partial$^*$ | Partial$^*$ | Yes | No | Yes$^\S$ |
| # Fidelity Params | 4 | 1 | 1 | 1 | 1 | - | 3$^\S$ |
| Multi-Objective? | Yes | Yes | Partial | Yes | Yes | Yes | Yes |
| Surrogate? | Yes | No | Yes | No | No | Yes | No |

Table 7: A comparison of NAS-HPO-Bench-II, JAHS-Bench-201 and NAS-Bench-201. (∗) denotes that full information on the number of seeds for all datasets is not available for NAS-Bench-201, so we indicate the number of evaluations on CIFAR-10 only.

| Property | NAS-HPO-Bench-II | NAS-Bench-201 | JAHS-Bench-201 |
|---|---|---|---|
| Cell Space size | 4,096 | 15,625 | 15,625 |
| Batch Size | 6 values | - | - |
| Learning Rate | 8 values | - | Continuous |
| Weight Decay | - | - | Continuous |
| Trivial Augment | - | - | On/Off |
| Fidelities | - | - | Epochs, N, W, R |
| # Tasks | 1 | 3 | 3 |
| # Trained configs | 4.8K | 32.8K$^*$ | 810K |

and NAS-Bench-HPO-II, JAHS-Bench-201 covers a larger space in both the architectural and hyperparameters axes, with some of the hyperparameters being continuous, for which we use a surrogate model to model the space. Moreover, we store 20 metrics in our evaluations, more than any other previous benchmark. By storing the metrics' values at each epoch we enable full learning curve support, and not only partial on a handful of epochs like NAS-HPO-Bench-II does. JAHS-Bench-201 strictly extends NAS-Bench-201 [6], by adding continuous and discrete hyperparameter dimensions, as well as multiple fidelities, objectives and tasks. In Table 7, we provide the ranges for our space and compare it in more detail to NAS-HPO-Bench-II and NAS-Bench-201.

Most works in the benchmark literature support a single fidelity parameter, the number of epochs of training of a configuration [50, 6, 5], allowing a user to query either full or partial learning curves. NATS-Bench [40] approached multi-fidelity support in a novel way and implemented a parameter for network width, directly comparable with $W$ in our work, in a way that allows it to support up to $8^5$ unique values, in addition to the number of training epochs. However, this also splits their search space into two spaces with some overlap. In contrast, JAHS-Bench-201 is able to provide full support for all 4 of our fidelity parameters across our entire joint NAS and HPO search space.

Most benchmarks in the literature record multiple metrics for their configurations' performance [40, 42, 50], which all have the potential to support multi-objective optimization algorithms. However, supporting multiple metrics does not necessarily translate into effective support for multi-objective optimization, as some metrics may completely subsume each other. For instance, NAS-HPO-Bench-II only provides the training and validation accuracies, which tend to be highly correlated,

and are not typically the scope of multi-objective optimization. Conversely, HW-NAS-Bench [46] demonstrates innovative and effective use of true multi-objective optimization for the NAS-Bench-201 and FBNet [52] contexts.

## 6 Conclusion and discussion

Recent work in the Neural Architecture Search (NAS) community [10] and our own experiments call for a shift in focus towards Joint Architecture and Hyperparameter Search (JAHS). To facilitate this shift, we provide a collection of cheap-to-evaluate surrogate benchmarks, JAHS-Bench-201, that support a broad range of algorithmic frameworks and optimization tasks, including multi-objective, cost-aware, multi-fidelity, and multi multi-fidelity optimization.

While we expect JAHS-Bench-201 to contribute to fast progress on JAHS, there are limitations to it and our methodology. Firstly, as JAHS-Bench-201 extends NAS-Bench-201[6] with a fidelity and hyperparameter space, we inherit its scope for included architectures. We chose this architecture search space primarily due to the popularity it enjoys in benchmarking for NAS, thereby enabling NAS researchers who wish to work on JAHS to have a familiar starting point. For a more holistic evaluation of JAHS algorithms, the research community should explore additional search spaces. Secondly, while we collected many potential objectives, a large number of them are highly correlated (see Figure 5), leading to clusters of similar metrics and limited options to create multi-objective tasks. Thirdly, due to the breadth of different types of algorithmic settings we support with our 24 leaderboards, we do not consider a wide array of algorithms for each of these in our experiments and initial leaderboard entries, but we encourage users of our benchmarks to develop JAHS algorithms for their specific setting of interest to populate the leaderboards. Fourthly, as we detail further in Appendix F, there is some noise in the performance data for the Colorectal-Histology task. Our surrogate models for this task perform well given the circumstances, but still underperform as compared to those for CIFAR-10 and Fashion-MNIST. Finally, larger hyperparameter search spaces than the one we considered could prove very useful, but were not feasible for our work since every added hyperparameter adds to the burden of compute requirements. Thus, we leave extending our search space to future work.

## Acknowledgements and disclosure of funding

Robert Bosch GmbH is acknowledged for financial support. We gratefully acknowledge support by BMBF grant DeToL, by the Deutsche Forschungsgemeinschaft (DFG, German Research Foundation) under grant number 417962828, by TAILOR, a project funded by the EU Horizon 2020 research and innovation programme under GA No 952215, and by European Research Council (ERC) Consolidator Grant "Deep Learning 2.0" (grant no. 101045765). Funded by the European Union. Views and opinions expressed are however those of the author(s) only and do not necessarily reflect those of the European Union or the ERC. Neither the European Union nor the ERC can be held responsible for them. The authors gratefully acknowledge the Gauss Centre for Supercomputing e.V. (`www.gauss-centre.eu`) for funding this project by providing computing time on the GCS Supercomputer SuperMUC-NG at Leibniz Supercomputing Centre (`www.lrz.de`).



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
