# OpenReview forum: "JAHS-Bench-201: A Foundation For Research On Joint Architecture And Hyperparameter Search"
_NeurIPS.cc/2022/Track/Datasets_and_Benchmarks — NeurIPS 2022 Datasets and Benchmarks _

### Official Review · Reviewer_H92w · 2022-07-13
**The next step in NAS**

**Rating:** 9
**Confidence:** 4

**Strengths:**

The dataset is very extensive and structured. Research in NAS is a developing field and this benchmark will provide the community with an easy access point.

**Weaknesses:**

A weakness is the limited search space for architectures to be considered.

**Additional Feedback:**

There was a small typo in line 73: hyperparametwe

**Clarity:**

This paper is very well written. The necessary information is provided in a structured and clear way. Tables that are provided are very helpful l and support the reader.

**Correctness:**

The dataset is constructed in a sound way. The authors rely on highly populair previous work in the field.

**Documentation:**

The benchmark and code is easily accessible via GitHub and well documented on the page. The licensing and hosting is all laid out. I could not find a specific reference to a maintenance plan.

**Ethics:**

there are none.

**Relation To Prior Work:**

The authors clearly discuss the connection and differences compared to the NAS-Bench-201 benchmark, which they use as a starting point.

**Summary And Contributions:**

This paper provides the information of the construction of a new benchmark set, JAHS-bench-201. This benchmark includes the results of Neural Architecture search combined with hyper-parameter optimisation. These two processes are both hard and time-consuming. This benchmark will result in making research in the combination of the two optimisations less time and energy consuming for the testing of future methods in this field. This is the main contribution.

---

> ### Author Response · Authors · 2022-08-14
> **Initial reply**
>
> Thank you very much for calling JAHS the next step in NAS! We agree and hope that our work will play a pivotal role in the further development of this field. We also want to thank you for your very clear statement on our main contribution in “Summary And Contributions” that captures the potential impact of JAHS-Bench-201 on the research community strikingly well. In the following, we would like to discuss the concerns you raised in your review.
>
> > A weakness is the limited search space for architectures to be considered.
>
> We do agree and have stated this as a limitation in Section 6. We have now added our reason for choosing the NAS-Bench-201 space to the discussion in Section 6 and also below.
>
>  While the NAS-Bench-201 architecture search space is limited, it is very popular in the field and therefore serves as a familiar starting point for NAS researchers that want to work on JAHS.
>
> > The benchmark and code is easily accessible via GitHub and well documented on the page. The licensing and hosting is all laid out. I could not find a specific reference to a maintenance plan.
>
> Thanks for your detailed assessment on our documentation! As the maintenance plan is an important aspect, we now feature our statement on it more prominently than we had in our initial submission. We now point towards our statement on the maintenance plan from Appendix B.
>
> Details on our maintenance can be found in our repository’s documentation at: https://automl.github.io/jahs_bench_201/performance_dataset/. In essence, the current hosting solution is an interim solution for the purposes of the review process. We will shift our data to Figshare, which is a trusted service in the scientific community for making scientific data publicly available for the long-term.
>
> In conclusion, we really appreciate your high opinion of our work and that you invested your time into reviewing and suggesting further improvements. We would be very interested in any additional feedback you may have and whether our improvements and replies above can justify going the last step in terms of review score.

---

### Official Review · Reviewer_D7L6 · 2022-07-21
**Extremely thorough and clear paper presenting a novel and useful dataset**

**Rating:** 9
**Confidence:** 4

**Strengths:**

The paper is extremely clear with a strong justification for the need of the dataset and strong ties to previous work. The dataset itself is extremely well documented with examples, easily accessible data, and built-in leaderboards. In addition to presenting the dataset, the authors conduct a thorough investigation into the usefulness of the dataset and JAHS more broadly. Their analysis seems sound and provides a strong justification for the novelty of this contribution.


**Weaknesses:**

This is a very strong submission. The authors clearly outline the significance of the contribution, relevance to the broader research community, accessibility and accountability, and ethical and social implications. Minor revisions are needed for clarity and some questions are presented in the additional feedback section for consideration.

**Additional Feedback:**

Why was batch size not used as a hyperparameter as in NAS-HPO-Bench-II?
Why was data after divergence excluded?
And why weren’t divergent configurations used in surrogate models?


**Clarity:**

Yes. Some typos listed below:

Line 73: misspelling: "hyperparametwe"
Line 146: Inconsistent use of “(multi) multi-fidelity,” “\textit{multi} multi-fidelity,” and “multi multi-fidelity.”
Line 197: “​700 larger” should be “700x larger”


**Correctness:**

Yes, to the best of my knowledge.


**Documentation:**

Yes, the documentation is thorough and clear.

**Relation To Prior Work:**

Yes, the paper provides an extensive outline of the differences between JAHS-Bench-201 and prior NAS and HPO benchmarks.


**Summary And Contributions:**

This paper presents JAHS-Bench-201, an extension of the popular neural architecture search benchmark NAS-Bench-201. JAHS-Bench-201 is a benchmark for testing joint architecture and hyperparameter search algorithms. In addition to presenting the dataset, this paper justifies its contributions by presenting promising joint architecture and hyperparameter search results.

---

> ### Author Response · Authors · 2022-08-14
> **Initial reply**
>
> Thank you for the strong appreciation of our work, including the relevance of JAHS to the community, our thorough empirical investigation, our benchmark design, and its documentation and data accessibility! Below, we directly address your raised concerns and questions.
>
> > Why was batch size not used as a hyperparameter as in NAS-HPO-Bench-II?
>
> Thanks for this question! We now address this in our discussion on limitations.
>
> The primary reason for doing so is the computational cost involved in constructing our benchmark, which increases for every new hyperparameter we include in our search space. Our dataset is already the largest such dataset in the public domain and each new hyperparameter included in the search space would require even more samples to be drawn, trained and evaluated in order to maintain the quality of our work. Further, we assessed the hyperparameters that we do include in JAHS-Bench-201 as more interesting.
>
>
> > Why was data after divergence excluded? And why weren’t divergent configurations used in surrogate models?
>
> The choice to exclude divergent configurations was made in order to maintain the simplicity of the surrogate model. XGBoost is not capable of handling NaN values. We are currently investigating some changes to the way our surrogate models work in order to potentially incorporate divergence prediction.
>
>
> > Typos
>
> Thank you for pointing these typos out to us! We have now corrected all of them.
>
> As a closing note, we very much appreciate the time you took to read and review our paper. Hopefully, we adequately address your remaining concerns about our work. Even though we already received an excellent rating, we would be glad to hear more about where our paper falls short of taking that final step.

---

### Official Review · Reviewer_9zBC · 2022-07-27
**A more comprehensive benchmark for joint architecture and hyperparameter search**

**Rating:** 7
**Confidence:** 3

**Strengths:**

- The benchmark focuses on joint architecture and hyperparameter search which is required by many real-world AutoML applications nowadays. Hence, the topic studied is well-motivated and has practical significance.

- The benchmark proposes a much more comprehensive search space comprised of an architecture space, a hyperparameter space, and a fidelity parameter space. In addition to the novel design of the search space, the benchmark considers a wide range of optimization tasks and algorithmic settings including multi-fidelity optimization, which distinguishes it from previous similar work. The considered data and tasks are also diverse, thus the benchmark can reflect better on the actual performance of different network models.

- This paper answers interesting research questions such as how important it is to jointly optimize in the space of architectures and hyperparameters. It provides valuable insights into possible research directions in AutoML.

- The paper is well-organized and well-written. The code, datasets, and experiment scripts are provided for reproducibility. Trails are over 10 random seeds, and both mean and standard error are provided, so the results seem credible.

**Weaknesses:**

- Jointly optimizing neural network architectures and training hyperparameters is not a new topic for benchmark work. As mentioned in the paper, previous work such as NAS-HPO-Bench and NAS-HPO-Bench-II studies the same problem, so the novelty of this work mainly lies in the extended search space and more diverse evaluation settings.

- The hyperparameter search space is somehow restricted as some commonly considered hyperparameters are not considered, such as optimizer type, momentum, batch size, etc. How easy it is to extend the set of considered hyperparameters?

- It may be beneficial to include instructions on testing a custom NAS/JAHS algorithm on the search space in the code repo.

- To answer the first question in section 4, the paper compares the performance of randomly sampled architectures and hyperparameter configurations. However, rather than using only the random search baseline, it can be more informative to include comparisons between more sophisticated NAS algorithms and HPO algorithms.

**Additional Feedback:**

N/A

**Clarity:**

The paper is mostly well-written and also tries to be accessible to researchers working outside of the NAS domain.


**Correctness:**

The claims made in the paper appear to be correct and the experimental protocol seems reasonable.


**Documentation:**

Yes, the code is provided and the experiment details, including the search and evaluation hyperparameters, are provided.


**Ethics:**

There seem to be no immediate ethical concerns.


**Relation To Prior Work:**

The authors discuss related work in Section 5 and make it clear why previous NAS-HPO benchmarks are limited and insufficient.


**Summary And Contributions:**

This paper argues that hyperparameter optimization plays an equally important role as neural architecture search in getting well-performed models. Hence, the research community should focus more on developing methods that jointly optimize the architecture and training hyperparameters. To call for more attention to joint architecture and hyperparameter search (JAHS), the paper proposes a new benchmark, called JAHS-Bench-201, that evaluates 15k architectures across 3 image classification tasks with 4 different hyperparameters and 4 fidelity parameters. In addition to having an extensive search space, JAHS-Bench-201 also supports benchmarking in 6 optimization tasks and 4 algorithmic settings. Lastly, the paper provides insights into why JAHS matters and studies the efficacy of different fidelity parameters.

---

> ### Author Response · Authors · 2022-08-14
> **Initial reply**
>
> We appreciate you highlighting JAHS as well-motivated and having practical significance, and thank you for your compliments regarding the various design choices behind our benchmark, the utility of our experiments, and the overall quality of our work. In the following, we would like to directly address the concerns you raised in your review.
>
> > The hyperparameter search space is somehow restricted as some commonly considered hyperparameters are not considered, such as optimizer type, momentum, batch size, etc. How easy it is to extend the set of considered hyperparameters?
>
> Whereas we agree that it would be very useful to have a much larger hyperparameter search space, adding additional hyperparameters will necessitate evaluating more configurations in order to have a better coverage of this larger space. As our dataset is already the largest such dataset and required a very large amount of computational resources to create (6 million core-h), we leave yet larger search spaces to future work and have included this to our discussion on limitations.
>
> > It may be beneficial to include instructions on testing a custom NAS/JAHS algorithm on the search space in the code repo.
>
> This is an excellent suggestion! We have added an example for running random search using the benchmark with additional instructions (https://github.com/automl/jahs_bench_201_experiments/blob/master/jahs_bench_201_experiments/tasks/run_example.py).
>
> > To answer the first question in section 4, the paper compares the performance of randomly sampled architectures and hyperparameter configurations. However, rather than using only the random search baseline, it can be more informative to include comparisons between more sophisticated NAS algorithms and HPO algorithms.
>
> We agree and, while RS already provides important insights, we have now added Bayesian Optimization versions for all our evaluation settings and included figures in our paper (Figure 2 and Figure 4).
>
> > Jointly optimizing neural network architectures and training hyperparameters is not a new topic for benchmark work. [...]
> > The authors [...] make it clear why previous NAS-HPO benchmarks are limited and insufficient.
>
> We agree that JAHS by itself is not a novel topic for benchmarking. That said, as you point out yourself in the second statement above, we make it clear that there are only a very small number of existing benchmarks on this topic and they all exhibit a number of limitations which render them insufficient to provide a foundation of research on JAHS the way JAHS-Bench-201 does.
>
> Finally, thank you very much for taking the time to review our paper and providing us with valuable feedback, prompting several improvements to our submission. We hope we have clarified some of your concerns and would appreciate it if you consider improving the rating in your review. We welcome any further suggestions or feedback that you would be willing to share.

---

> > ### Comment · Reviewer_9zBC · 2022-08-26
> > **Thank you for the response**
> >
> > Thank you for the additional comments. I'll keep my score as it is.

---

### Official Review · Reviewer_Ho14 · 2022-07-28
**Practical and Useful Benchmark for the NAS Community**

**Rating:** 7
**Confidence:** 4

**Strengths:**

&nbsp;

1. The idea that hyperparameters and architectures should be considered jointly when evaluating hyperparameter optimisation algorithms is a very logical one and the proposed benchmark saves HPO researchers a great deal of time in terms of constructing a suitable evaluation framework.

&nbsp;

**Weaknesses:**

&nbsp;

## **MAJOR POINTS** ##

&nbsp;

1. The statement in the introduction,

&nbsp;

> Despite recent work showing that good hyperparameter settings are even more important than using the best architecture.

&nbsp;

seems to contradict the results of the experiment provided in Figure 1 i.e. NAS-only optimisation achieves lower error relative to HPO-only optimisation. It would be great if the authors could clarify this?

&nbsp;

2. As the reviewer understands, the difficulty in extending tabular NAS benchmarks to consider continuous hyperparameter domains jointly is that the hyperparameter domains are continuous and hence every possible configuration cannot be computed. As such, the authors interpolate a (presumably dense) grid of the continuous hyperparameter space using a surrogate model. I'm slightly confused about the methodology used to assess the model fit on this grid using the train/validation/test split in Appendix D. Presumably the test set is structured such that it contains points such that the train set becomes a finer grid relative to the entire dataset i.e. the test points lie inbetween the data points constituting the grid of the training points? Is this what the authors mean when they describe the construction of the splits in terms of fidelities? Presumably the surrogate model is then retrained on the entire set of collected data?

&nbsp;

3. Why is the R^2 for the surrogate on the Colorectal-histology task so low?

&nbsp;

## **MINOR POINTS** ##

&nbsp;

1. It would be great if the references appeared in numbered order e.g. via the LaTeX command:

&nbsp;

> \bibliographystyle{unsrtnat}

> \setcitestyle{numbers,open={[},close={]},citesep={,}}

&nbsp;

2. Typo line 29, "fueled"

&nbsp;

3. Line 42, the nested (multiple) is confusing.

&nbsp;

4. It may be worth making the distinction between an architecture and a hyperparameter (e.g. activation function type) in the introducton to make the paper more accessible to readers who are unfamiliar with the NAS literature.

&nbsp;

5. Line 53, typo, full stop in place of comma in 270,000.

&nbsp;

6. The attempt to run the benchmark using the minimal example provided in the repository resulted in an error related to XGBoost.

&nbsp;

7. Line 56, typo, capitalisation of "The" after the colon.

&nbsp;

8. Line 63, typo, superfluous colon.

&nbsp;

9. In Table 4 of the SI perhaps it would be an idea to give time in a more intuitive unit such as days.

&nbsp;

10. Line 73, typo in hyperparameter.

&nbsp;

11. Line 105, typo, missing comma in 10,000.

&nbsp;

12. R^2 instead of R2 for the coefficient of determination.

&nbsp;

13. The NAS and HPO acronyms are defined in the introduction. It would be great if they could be used consistently in the rest of the paper.

&nbsp;

14. It might be worth moving some of the details from section D of the appendix into the main paper. The page limit for the submission is actually 9 pages so this should be achievable.

&nbsp;

15. Line 133, typo, "settings".

&nbsp;

16. Line 135, typo, capitalisation following colon.

&nbsp;

17. Line 137, typo, leaderboards.

&nbsp;

18. Line 155, Might be worth using JAHS as the acronym since it has already been defined.

&nbsp;

19. The number of trials over which the standard error is computed could be given directly in the captions of Figure 1 and Figure 2.

&nbsp;

20. In terms of HPO benchmarks it may also be worth referencing Bayesmark and the NeurIPS black-box optimisation challenge benchmarks [1,2].

&nbsp;

21. Line 189, typo "hardware".

&nbsp;

22. The 6\dot\mathbb{R}^2 is somewhat confusing in Table 6.

&nbsp;

23. Line 197, typo, "700 times larger".

&nbsp;

24. Line 217, typo, "multi" instead of "multy".

&nbsp;

25. Line 41, Appendix D, missing "so".

&nbsp;

26. Typos in capitalisation in the references, e.g. "Bayesian".

&nbsp;

27. Line 82, Appendix G, missing space when mentioning "Table 4".

&nbsp;

28. Table 4 caption, missing "the".

&nbsp;

## **REFERENCES** ##

&nbsp;

[1] Turner et al. Bayesian Optimization is Superior to Random Search for Machine Learning Hyperparameter Tuning: Analysis of the Black-box Optimization Challenge 2020. In NeurIPS 2020 Competition and Demonstration Track, 2021.

&nbsp;

[2] Cowen-Rivers et al. HEBO: Pushing The Limits of Sample-Efficient Hyper-parameter Optimisation. Journal of Artificial Intelligence Research, 2022.

&nbsp;

**Additional Feedback:**

&nbsp;
All feedback provided in the main response.
&nbsp;

**Clarity:**

&nbsp;
cf. the main response.
&nbsp;

**Correctness:**

&nbsp;
cf. the main response.
&nbsp;

**Documentation:**

&nbsp;
Adequately addressed. Licensing plan included. All issues related to reproducibility issues mentioned explicitly in the submission.
&nbsp;

**Ethics:**

&nbsp;
Adequately addressed.
&nbsp;

**Relation To Prior Work:**

&nbsp;
Adequately addressed.
&nbsp;

**Summary And Contributions:**

&nbsp;

The authors extend the set of NAS benchmarks by introducing a new benchmark focussed on joint architecture and hyperparameter search. Currently, I lean towards acceptance and will consider raising my score following the authors' response to the points raised below.

&nbsp;

---

> ### Author Response · Authors · 2022-08-14
> **Initial reply 1/2**
>
> We are glad that you share our appreciation of JAHS as well as the value of a comprehensive benchmark to researchers in the field. Your review was very helpful and allowed us to improve our submission in several ways, thanks! Below, we address the concerns you raised, list the resulting improvements, and request your feedback on the same.
>
> > [...] “Despite recent work showing good hyperparameter settings are even more important than using the best architecture.” seems to contradict the results of the experiment provided in Figure 1 i.e. NAS-only optimisation achieves lower error relative to HPO-only optimisation.
>
> We agree and have changed this sentence to “Despite recent work showing good hyperparameter settings can be more important than using the best architecture.”
>
> > [...] I'm slightly confused about the methodology used to assess the model fit on this grid using the train/validation/test split in Appendix D. Presumably the test set is structured such that it contains points such that the train set becomes a finer grid relative to the entire dataset i.e. the test points lie inbetween the data points constituting the grid of the training points? [...]
>
> It appears to us that there has been a misunderstanding regarding how to interpret the contents of Appendix D. As such, we have made changes to Appendix D in order to make it more explicit and unambiguous. We do not perform grid sampling, as all of our configurations are sampled randomly from the search space. When designing the train, validation and test splits, a naive approach of a 70-15-15 split would not be appropriate for our data since it contains time series data - all data points from multiple epochs of training and evaluation of the same configuration belong to a single time series. As such, they needed to be grouped together as a single data point for the train, validation and test splits. Also, we wanted to avoid introducing sampling biases in terms of the number of grouped data points from each fidelity group that are assigned to each split. Thus, rather than the test set being designed to interpolate between the train set points, the train, validation and test sets are all designed to retain the statistical properties of the overall performance dataset as well as remain statistically independent of each other.
> We would be happy to hear your feedback on this issue, whether this explanation clarifies this detail, and whether our changes to Appendix D sufficiently resolve any ambiguities. If not, we would appreciate your advice on what could be improved further.
>
> > Why is the R^2 for the surrogate on the Colorectal-histology task so low?
>
> Thank you for bringing this topic up! We are currently running some tests and analyses on this.
>
> > It would be great if the references appeared in numbered order
>
> We agree! Following your advice, we have updated our numbering scheme.
>
> > In terms of HPO benchmarks it may also be worth referencing Bayesmark and the NeurIPS black-box optimisation challenge benchmarks [1,2].
>
> Agree, done!
>
> > In Table 4 of the SI perhaps it would be an idea to give time in a more intuitive unit such as days.
>
> Agreed, we have updated that table accordingly.

---

> > ### Author Response · Authors · 2022-08-14
> > **Initial reply 2/2**
> >
> > > The NAS and HPO acronyms are defined in the introduction. It would be great if they could be used consistently in the rest of the paper. [...] Line 155, Might be worth using JAHS as the acronym since it has already been defined.
> >
> > This is a fair point. We made the choice to not use these acronyms in certain places consciously in order to accommodate skimming readers who may not be very familiar with the relevant literature and would otherwise be forced to repeatedly return to previous sections of the paper in order to decrypt the acronyms. That said, we understand that consistent usage of the acronyms would make for a smoother reading experience for many readers and welcome any further opinions on this style choice.
> >
> > > The attempt to run the benchmark using the minimal example provided in the repository resulted in an error related to XGBoost.
> >
> > Thank you for pointing out this issue to us. We would be glad to know more details about this as well as the environment you use, in order to debug this. For example, would you happen to be using MacOS? We discovered that XGBoost does not enjoy the same level of support for MacOS as it does for Linux-based OS’. Since all of our previous internal testing was performed on various Linux-based OS’, we will need to dedicate more time and effort towards investigating the situation for MacOS. We have now added a warning for Mac users in our repository’s Readme.
> >
> > > The number of trials over which the standard error is computed could be given directly in the captions of Figure 1 and Figure 2.
> >
> > We agree and have updated the relevant captions.
> >
> > > Typos
> >
> > Thank you very much for the painstaking effort involved in this! We have addressed all of the typos mentioned by you.
> >
> > As a final note, thank you very much for your very detailed review and great suggestions! We believe the resulting changes have improved the quality of our paper. As you mentioned you will consider raising your score, we would be happy to hear your assessment of our response and the improvements to our submission! We also welcome any further suggestions or critiques you may have for our paper.

---

> > > ### Comment · Reviewer_Ho14 · 2022-08-24
> > > **Many Thanks for the Response**
> > >
> > > I thank the authors for their response, for clarifying the contents of Appendix D, and for implementing changes to the paper. I take the authors' point that the consistent use of acronyms is a matter of taste and indeed, it may be beneficial for some readers to have reminders throughout the paper!
> > >
> > > &nbsp;
> > >
> > > Indeed MacOS would appear to be the problem!
> > >
> > > &nbsp;
> > >
> > > I look forward to the authors' response regarding the colorectal-histology task. In the meantime I am happy to raise my score.

---

### Official Review · Reviewer_Xwu4 · 2022-07-28
**Solid contribution but slight concerns about the lack of details w.r.t. surrogates**

**Rating:** 8
**Confidence:** 5

**Strengths:**

- This work evaluates many dimensions of the JAHS problem, with plenty of metrics, support for multi-fidelity optimization
- Reasonable choices of hyperparameters
- Builds on NAS-Bench-201, a widely adopted tabular NAS search space.
- Additional hyperparameters controlling network capactity -- these are often neglected in tabular NAS-Benches.
- The idea to allow the image scaling to be a hyperparameter is interesting.
- The authors use their benchmark to evaluate specific scientific hypotheses about the joint optimization of architectures and hyperparameters as well as a comparison of different types of fidelities. Both of which are understudied in the literature.
- The multi-multi-fidelity setting is interesting and is a promising direction for future work promoted by this benchmark.

**Weaknesses:**

- The use of surrogate models to generate the performance predictions is potentially risky. As far as I'm aware, using surrogates to produce benchmarks via performance prediction using an XGBoost regressor has mainly been used (somewhat) in NAS benchmarks, and has not been thoroughly vetted for hyperparameter benchmarks. The authors report some reasonable numbers in the main text that support the usage of surrogates, but this is a point that should have more evidence/support provided in the main text.
- While the multi-fidelity evaluation in Figure 2 is interesting in that many of the fidelity types perform similarly, a separate evaluation combining multiple fidelities at once seems to be missing. This setting is described as future work, but this experiment would also help to strengthen the current claims in the "Are some fidelities more useful than others?" section.
- Given that surrogate performance predictors were employed to generate the benchmark, it would have been nice to include more datasets beyond the three currently included in the benchmark.
- Additional experiments, scientific hypothesis enabled by the benchmark, or any additional content would help to improve the paper. The current version of the paper has 8 pages of content, though the page limit is 9 pages.

**Additional Feedback:**

- The page limit for this track is 9 pages -- though the paper has 8 pages of content.
- (Line 73) typo "hyperparametwe"

**Clarity:**

- The paper is written clearly enough.

**Correctness:**

- Everything reported in the paper seems correct, although the surrogate modeling concerns me.

**Documentation:**

- Quite thorough documentation is provided on GitHub and the associated leaderboard website.

**Ethics:**

- No issues.

**Relation To Prior Work:**

- Prior work is described in detail, and the relationship between JAHS-Bench-201 and other Benchmarks is discussed.

**Summary And Contributions:**

This paper contributes a tabular benchmark called JAHS-Bench-201, which measures the performance configurations in a joint hyperparameter and NAS search space over three image classification tasks: CIFAR-10, Colorectal-Histology, and Fashion-MNIST. There are 140 million reported configurations per dataset, most of which are derived from surrogate predictions learned from an initial set of 270,000 evaluated configurations on each task. Furthermore, of the 270,000 configurations evaluated on each dataset, 20 different performance metrics are captured.

---

> ### Author Response · Authors · 2022-08-14
> **Initial reply 1/2**
>
> Thank you for stating our work being a solid contribution and highlighting the research topics our benchmark promotes as understudied and interesting. Below we explain the improvements to our paper in response to your review and address concerns you raised, we would like to start with a misunderstanding about our work expressed in your summary:
>
> > This paper contributes a tabular benchmark called JAHS-Bench-201 [...] There are 140 million reported configurations per dataset, most of which are derived from surrogate predictions learned from an initial set of 270,000 evaluated configurations on each task.
>
> The surrogate models were not used to generate any of the 140 million data points; rather, these data points were used to train a surrogate model that can then be queried instead of training from scratch. Therefore, as we state in the abstract, “we present JAHS-Bench-201, the first collection of surrogate benchmarks for JAHS, [...]”, not a tabular benchmark. The benefit of surrogate benchmarks over tabular benchmarks is that we can support continuous parameters (e.g., learning rate) and make large spaces (such as ours) feasible. We have now updated Section 2.2 to be clearer about this. We also welcome any updates to your initial review in light of this.
>
> > Given that surrogate performance predictors were employed to generate the benchmark, it would have been nice to include more datasets beyond the three currently included in the benchmark.
>
> We do agree that more datasets would be beneficial, but as we clarified in the response above, performance predictors were not used to generate the benchmark. Rather, a large amount of compute (in our case 6 million core-h) was required to generate our performance dataset to then fit the surrogate models. As creating the surrogate benchmarks for 3 datasets already required a vast amount of compute power (making our performance data the most extensive such dataset ever collected in the public domain), doing this for another dataset would be infeasible in the short time of the rebuttal.
>
>
> > The use of surrogate models to generate the performance predictions is potentially risky. As far as I'm aware, using surrogates to produce benchmarks via performance prediction using an XGBoost regressor has mainly been used (somewhat) in NAS benchmarks, and has not been thoroughly vetted for hyperparameter benchmarks.
>
> Surrogate benchmarks have been used frequently in NAS (forerunners include NAS-Bench-301 [1] and NAS-Bench-x11 [2]), have a long history in HPO [3], and surrogate models based on gradient boosting have been thoroughly vetted for both NAS [4] and HPO [3]. We have improved Section 2.3 to clarify that such surrogate models have been vetted for both NAS and HPO benchmarks.
>
> [1] J. Siems, L. Zimmer, A. Zela, J. Lukasik, M. Keuper, and F. Hutter. NAS-bench-301 and the case for surrogate benchmarks for Neural Architecture Search. arXiv:2008.09777v4 [cs.LG], 2020.
>
> [2] S. Yan, C. White, Y. Savani, and F. Hutter. NAS-Bench-x11 and the power of learning curves. Advances in Neural Information Processing Systems, 34, 2021.
>
> [3] K. Eggensperger, F. Hutter, H.H. Hoos, and K. Leyton-Brown. Efficient benchmarking of hyperparameter optimizers via surrogates. In B. Bonet and S. Koenig, editors, Proceedings of the Twenty-ninth National Conference on Artificial Intelligence (AAAI’15), pages 1114–1120. AAAI Press, 2015.
>
> [4] Y. Mehta, C. White, A. Zela, A. Krishnakumar, G. Zabergja, S. Moradian, M. Safari, K. Yu, and F. Hutter. NAS-Bench-Suite: NAS evaluation is (now) surprisingly easy. In Proceedings of the International Conference on Learning Representations (ICLR’22), 2022.
>
>
> > The authors report some reasonable numbers in the main text that support the usage of surrogates, but this is a point that should have more evidence/support provided in the main text.
>
> We agree that the evidence for the validity of our surrogate models should be more prominently featured in the main text of our paper. Accordingly, we have moved Table 4 to Section 2.3 in the paper. Thanks!

---

> > ### Author Response · Authors · 2022-08-14
> > **Initial reply 2/2**
> >
> >
> > > While the multi-fidelity evaluation in Figure 2 is interesting in that many of the fidelity types perform similarly, a separate evaluation combining multiple fidelities at once seems to be missing. This setting is described as future work, but this experiment would also help to strengthen the current claims in the "Are some fidelities more useful than others?" section.
> >
> > Thanks for this suggestion! We agree and have added simple multi multi-fidelity algorithms that traverse on the diagonal of the fidelity hypercube using Hyperband (see Figure 3 and 4 in the revised paper).
> >
> > > Additional experiments, scientific hypothesis enabled by the benchmark, or any additional content would help to improve the paper.
> >
> > We agree and, among other improvements and new content, have added:
> > * Experiments using multi multi-fidelity algorithms
> > * Experiments using Bayesian Optimization
> > * A research question on the correlation of objectives
> >
> > We hope we have adequately addressed your concerns and clarified any misunderstandings regarding the role surrogate models play in our benchmark, and, if so, we would appreciate it if you would consider updating your score. We would also welcome any further suggestions you may have for improving our paper.

---

> > > ### Comment · Reviewer_Xwu4 · 2022-08-15
> > > **Thanks! I have updated my score.**
> > >
> > > Thank you for the clarifications and for the additional content, which I think has strengthened the paper considerably. Seeing as all of my concerns have been adequately (actually thoroughly) addressed, I will update my score.

---

> > > > ### Author Response · Authors · 2022-08-29
> > > > **Follow-up**
> > > >
> > > > > Indeed MacOS would appear to be the problem!
> > > >
> > > > We will open an issue for xgboost and will continue pushing for a fix. Thanks for bringing this issue to light: as xgboost is a common dependency for surrogate benchmarks, a fix will improve accessibility of HPO/NAS/JAHS benchmarks beyond our own work.
> > > >
> > > > > [...] Authors' response regarding the colorectal-histology task.
> > > >
> > > > We performed a number of experiments to investigate the surrogate model performance on Colorectal-Histology; our conclusion is:
> > > >
> > > > - Task-dependent metrics, like accuracy, exhibit higher noise on Colorectal-Histology than on our other tasks. Taking this noise into account, the performance of the surrogate models is actually quite good. We have now added Appendix F, which contains the details of this analysis.
> > > > - For hardware dependent metrics, such as FLOPS and runtime, the lower correlations seem to stem from the performance data for Colorectal-Histology being collected in two batches with a few months gap in between and some performance characteristics out of our control seemingly changing across the two batches. This does not affect the other two tasks. We provide more details in Appendix F.
> > > >
> > > > We have added a discussion on the Colorectal-Histology data to our conclusion section.

---

### Author Response · Authors · 2022-08-14
**Changes to the Paper and Code**

To help simplify the review process, we list the most important improvements to our paper and code below.
* Clarifications to the surrogate creation and evaluation methodology.
* New experiments on multi multi-fideilty.
* New experiments using Bayesian Optimization
* New discussion points on the limitations of JAHS-Bench-201
* New code example showcasing how to run an algorithm against JAHS-Bench-201

Also, in our paper, we have highlighted important improvements to text (but not figures/tables to keep visual clarity) in yellow.

---

### Meta-Review · Area_Chair_VZk9 · 2022-09-09

**Recommendation:** Accept
**Confidence:** 5

**Metareview:**

The JAHS-Bench-201 provides a collection of surrogate benchmarks for JAHS that will enable research on multi-objective, cost-aware and multi-fidelity optimization methods. There is consensus among the reviewers that the contribution is significant and the discussion/rebuttal reinforces the strength of the paper.

---

### Decision · Program_Chairs · 2022-09-16

Accept